# COVID-19 vaccine hesitancy among non-refugees and refugees in Kenya

**Ryan T. Rego**[1]*, **Anthony K. Ngugi**[2], **Antonia Johanna Sophie Delius**[3], **Stanley Luchters**[2], **Joseph C. Kolars**[1], **Furqan B. Irfan**[4], **Eileen Weinheimer-Haus**[1], **Amina Abubakar**[5], **Reena Shah**[6], **Ji Zhu**[7], **Matthew L. Boulton**[8], **Timothy Hofer**[1‡], **Akbar K. Waljee**[1‡]

1 Center for Global Health Equity, Michigan Medicine, University of Michigan, Ann Arbor, Michigan, United States of America, 2 Dept. of Population Health, Aga Khan University, Nairobi, Kenya, 3 Poverty and Equity Global Practice, World Bank Group, Washington, District of Columbia, United States of America, 4 Institute of Global Health, Michigan State University, Lansing, Michigan, United States of America, 5 Institute for Human Development, Aga Khan University, Nairobi, Kenya, 6 Dept. of Internal Medicine, Aga Khan University, Nairobi, Kenya, 7 Dept. of Statistics, University of Michigan, LSA, Ann Arbor, Michigan, United States of America, 8 Dept. of Epidemiology, School of Public Health, University of Michigan, Ann Arbor, Michigan, United States of America

‡ TH and AKW are joint senior authors on this work.
* RegoR@UMich.edu

**Data Availability Statement:** The data used in this paper are available publicly from https://microdata.worldbank.org/index.php/catalog/3774.

## Abstract

Factors associated with COVID-19 vaccine hesitancy (which we define as refusal to be vaccinated when asked, resulting in delayed or non- vaccination) are poorly studied in sub-Saharan Africa and among refugees, particularly in Kenya. Using survey data from wave five (March to June 2021) of the Kenya Rapid Response Phone Survey (RRPS), a household survey representative of the population of Kenya, we estimated the self-reported rates and factors associated with vaccine hesitancy among non-refugees and refugees in Kenya. Non-refugee households were recruited through sampling of the 2015/16 Kenya Household Budget Survey and random digit dialing. Refugee households were recruited through random sampling of registered refugees. Binary response questions on misinformation and information were transformed into a scale. We performed a weighted (to be representative of the overall population of Kenya) multivariable logistic regression including interactions for refugee status, with the main outcome being if the respondent self-reported that they would not take the COVID-19 vaccine if available at no cost. We calculated the marginal effects of the various factors in the model. The weighted univariate analysis estimated that 18.0% of non-refugees and 7.0% of refugees surveyed in Kenya would not take the COVID-19 vaccine if offered at no cost. Adjusted, refugee status was associated with a -13.1[95%CI:-17.5,-8.7] percentage point difference (ppd) in vaccine hesitancy. For the both refugees and non-refugees, having education beyond the primary level, having symptoms of COVID-19, avoiding handshakes, and washing hands more often were also associated with a reduction in vaccine hesitancy. Also for both, having used the internet in the past three months was associated with a 8.1[1.4,14.7] ppd increase in vaccine hesitancy; and disagreeing that the government could be trusted in responding to COVID-19 was associated with a 25.9 [14.2,37.5]ppd increase in vaccine hesitancy. There were significant interactions between refugee status and some variables (geography, food security, trust in the Kenyan

**Funding:** Research reported in this publication was supported by the Office Of The Director, National Institutes Of Health (OD), the National Institute Of Biomedical Imaging And Bioengineering (NIBIB), the National Institute Of Mental Health (NIMH), and the Fogarty International Center (FIC) of the National Institutes of Health under award number U54TW012089 (Abubakar A and Waljee AK). The content is solely the responsibility of the authors and does not necessarily represent the official views of the National Institutes of Health.

**Competing interests:** The authors have declared that no competing interests exist.

government's response to COVID-19, knowing somebody with COVID-19, internet use, and TV ownership). These relationships between refugee status and certain variables suggest that programming between refugees and non-refugees be differentiated and specific to the contextual needs of each group.

## Introduction

People living in low- and middle-income countries (LMICs) face more than twice the risk of death if infected with COVID-19 than those living in high income countries (HICs) [1]. This disparity in mortality is in part due to differences in COVID-19 vaccine uptake, which can reduce the probability of death by 98% [2]. The contrast in vaccine receipt is stark; approximately 189 doses per 100 people have been administered in HICs while this figure is less than 100 doses per 100 people in LMICs [3]. At a regional scale, COVID-19 vaccine inequities are more evident still. In sub-Saharan Africa, less than 45 doses have been administered per 100 people, resulting in a much higher burden of preventable morbidity and mortality associated with COVID-19 infection [3]. Importantly, these inequities in vaccine uptake are not homogeneously distributed throughout populations throughout a given country. Specifically, refugees and other marginalized populations often experience lower vaccination rates relative to the general population [4]. As global supply chains increase availability of vaccinations, there is an increased onus on decreasing vaccination hesitancy. Vaccination hesitancy is poorly understood and a problem for both non-refugee and refugee populations, though particularly for refugee populations there are almost no studies on vaccination hesitancy for any vaccine, let alone the COVID-19 vaccine. As such, interventions among refugee populations are often modelled on those for non-refugee, despite a dearth of evidence to support similarities between the groups. A more tailored and targeted approach to addressing vaccine decision making should be pursued, given that factors in vaccine decision making likely vary substantially between refugee and non-refugee populations, with evidence needed to develop these [4].

Kenya is one of the most populous nations in sub-Saharan Africa with a population of more than 50 million, including over 500,000 refugees, making Kenya the country with the fourth highest number of refugees in sub-Saharan Africa. Greater than half of the refugees in Kenya are from Somalia, with the remainder in order of number from South Sudan (25%), the Democratic Republic of Congo (9%), Ethiopia (6%), Burundi (4%), and less than 5% from other countries. Slightly more than three-quarters of these refugees live in camp-based settings, with the remainder residing in urban areas. More than half of the refugees are under the age of 18, and there is an even split between males and females. Despite the large number of refugees in Kenya, little data exists on vaccine hesitancy in this vulnerable population. Slightly more information is available for the non-refugee population in Kenya: a recent study looking at only four counties (a mix of rural and urban) in Kenya found that 60% of respondents were vaccine hesitant, with associated factors being old age, low education, not adhering to government regulations on COVID, and not perceiving a risk of COVID-19, concerns with the vaccine, and religious beliefs [5]. However, as the data used in this study is only from four Kenyan counties, it cannot be generalized to the entirety of Kenya, and certainly cannot be generalized to Kenya's refugee population. Due to differences in lived experiences and access to healthcare, factors in vaccine hesitancy almost certainly differ between the refugee and non-refugee populations, particularly factors such as government trust and health literacy [4, 6]. To minimize unnecessary COVID-19 related morbidity and mortality among refugees in Kenya, who suffer increased risks of transmissions and related emergence of new variants due to high population

densities in their settlements, investigation into factors associated with vaccine hesitancy among refugee populations is urgently needed.

Using data collected between March and June of 2021, we examined factors associated with vaccine hesitancy among refugee and non-refugee populations in Kenya. The Kenyan government initiated a COVID vaccination program in March 2021 and aims to vaccinate at least 50% of the entire population by mid-2022 [5].

## Methods

Data were extracted from the 2021–2022 Rapid Response Phone Survey administered by the World Bank in collaboration with the United Nations High Commissioner for Refugees (UNHCR) and University of California, Berkeley [7, 8]. The survey was conducted in five rounds between May 2020 and June 2021, asking questions on participant demographics, beliefs and attitudes towards COVID-19, and socio-economic status. Only rounds four and five posed the question "Would you take the vaccine if offered to you at no cost," which was asked to the primary respondent who was initially contacted. This is along our definition of vaccine hesitancy, refusal to be vaccinated when asked, resulting in delayed or non- vaccination. To provide the most recent estimates and avoid having multiple responses from the same household, only round five observations were included (March to June 2021). Further information on the survey methodology has been published elsewhere [7, 8].

Key variables were selected and recoded as necessary, including creating an aggregate score of the misinformation and information questions (Table 1; S1 Appendix). Observations with missing data for key variables were excluded, including importantly observations with no responses for the variables addressing government trust which were asked randomly to a subset of the sample (accounting for 89% of the missingness). Variables of interest were broken down by refugee status (which we defined as refugee, for refugees, and non-refugees, for those who do not have refugee status, including Kenyan citizens and nationals) and vaccine hesitancy using an unadjusted analysis (not controlling for other factors) with household weights to make the sample representative of the current refugee and national number of households. An adjusted multivariate logistic regression (controlling factors for other factors in the regression with a yes/no binary outcome) was used to allow for comparisons between variables. Interaction terms were included to examine the relationship that refugee status and certain variables had on vaccine hesitancy, and non-significant interactions were excluded [9, 10]. A random forest model, looking for relationships through the construction of decision trees, was used to check for additional interactions. Results are presented as marginal effects [11].

## Results

### Unadjusted analysis

The round five dataset contained a total of 7,385 observations: 5,835 non-refugee and 1,550 refugee. We excluded 2,021 observations that did not include all variables of interest (1,732 non-refugee and 469 refugee). Remaining were 4,103 refugee observations, of which 737 (18.2% weighted to the overall Population of Kenya) were vaccine hesitant; and 1,081 refugees, of which 73 (7.0% weighted to the overall Population of Kenya) were vaccine hesitant (Table 1).

### Adjusted analysis

Fig 1 presents the marginal effects of each factor independently on vaccine hesitancy using the model specified in S2 Appendix and averaging over the distribution of the other variables in

**Table 1. Summary table presenting frequencies and rates of key variables, broken down by population and vaccine hesitancy** *n(weighted %).*

| | | Non-Refugee | | | Refugee | | |
|---|---|---|---|---|---|---|---|
| | | All (N = 4,103) | Not willing to get the vaccine (n = 737) | Willing to get the vaccine (n = 3,366) | All (N = 1,081) | Not willing to get the vaccine (n = 73) | Willing to get the vaccine (n = 1,008) |
| Demographics | Lives in an Urban Area | 2133 (37.3) | 365 (29.5) | 1768 (39.0) | 233 (15.4) | 20 (21.8) | 213 (14.9) |
| | Age *Mean (SD)* | 35.5 (12.0) | 33.8 (13.5) | 35.9 (11.6) | 35.8 (38.9) | 34.5 (41.5) | 35.9 (38.6) |
| | Gender (Female) | 2172 (47.4) | 385 (44.0) | 1787 (48.1) | 531 (52.0) | 36 (53.6) | 495 (51.9) |
| | Has Post-Primary Education | 2651 (68.3) | 401 (57.2) | 2250 (70.7) | 72 (7.9) | 6 (9.9) | 63 (7.7) |
| | Food Security | 1787 (45.2) | 384 (46.7) | 1403 (44.8) | 295 (29.7) | 5 (15.8) | 294 (30.6) |
| Behaviour and Exposure | **Do you think the government trustworthy in the way it manages the Coronavirus crisis** | | | | | | |
| | Disagree | 446 (13.3) | 126 (24.2) | 320 (10.9) | 102 (11.7) | 5 (8.2) | 97 (12.0) |
| | Neutral | 1221 (33.9) | 347 (51.5) | 874 (30.0) | 70 (7.2) | 6 (12.1) | 64 (6.8) |
| | Agree | 2436 (52.8) | 264 (24.5) | 2172 (59.1) | 909 (81.1) | 62 (78.8) | 847 (81.2) |
| | Knows somebody who has had covid | 299 (7.9) | 47 (6.9) | 252 (8.1) | 100 (10.5) | 5 (7.3) | 95 (10.7) |
| | Has had COVID-19 symptoms in the past 14 days | 929 (25.8) | 224 (30.7) | 705 (24.7) | 139 (13.8) | 5 (6.7) | 134 (14.3) |
| | Washes their hands more since COVID-19 began | 3852 (93.9) | 633 (81.7) | 3219 (96.6) | 1004 (92.0) | 66 (90.4) | 935 (92.2) |
| | Avoids contact with people more since COVID-19 began | 3974 (97.2) | 692 (94.5) | 3282 (97.8) | 1036 (95.5) | 70 (97.2) | 966 (95.3) |
| | Has gone shopping in the past 14 days | 2256 (60.4) | 416 (51.2) | 1840 (60.0) | 830 (76.0) | 49 (65.2) | 781 (77.9) |
| | Avoids groups of more than 10 more since COVID-19 began | 3351 (84.3) | 624 (87.9) | 2727 (83.4) | 1013 (93.1) | 68 (95.3) | 945 (93.0) |
| | Used the internet in the past 3 months | 2220 (57.4) | 417 (63.0) | 1803 (56.1) | 787 (69.2) | 46 (69.4) | 741 (69.2) |
| | Owns a radio | 3566 (86.7) | 629 (84.0) | 2937 (87.3) | 431 (41.6) | 15 (27.8) | 416 (42.6) |
| | Owns a television | 1893 (44.5) | 343 (47.5) | 1550 (43.8) | 247 (17.1) | 10 (8.8) | 237 (17.7) |
| Information/ Misinformation | Information Score *Mean (SD)* | 10.7 (0.8) | 10.6 (0.9) | 10.7 (0.8) | 10.4 (3.6) | 9.9 (3.9) | 10.5 (3.6) |
| | Misinformation Score *Mean (SD)* | 1.0 (1.3) | 0.9 (1.4) | 1.1 (1.3) | 1.5 (5.1) | 1.8 (5.1) | 1.5 (5.1) |

the joint population of non-refugees and refugees. Compared to non-refugees, refugee status was associated with lower vaccine hesitancy (-13.2[95%CI:-18.1,-8.2] percentage point difference (ppd)). Some variables had similar associations for both refugee and non-refugee, for which we did not include interactions: education beyond the primary level was associated with lower vaccine hesitancy (-12.5[-20.0,-5.2]ppd); as was washing hands more because of COVID-19 (-24.8[-33.4,-16.1]ppd); and avoiding handshakes more because of COVID-19 (-13.9[-25.5,-2.3]ppd). Non-significant interactions were seen for the other variables which did not include interactions (age, gender, going to the market, avoiding groups, radio ownership, and information/misinformation scores)

The adjusted model included several interactions of refugee status with the variables, for which we summarize the marginal probabilities (estimating rates of vaccine hesitancy averaged over the distribution of the other variables for the entire population) (Table 2). Refugees in urban settings had a higher marginal probability of vaccine hesitancy than their rural/camp-based counterparts, with the opposite for non-refugees. Further, the effect of trusting the government's response to COVID-19 differed between refugees and non-refugees: refugees who disagreed in trusting the government had a lower marginal probability of being vaccine

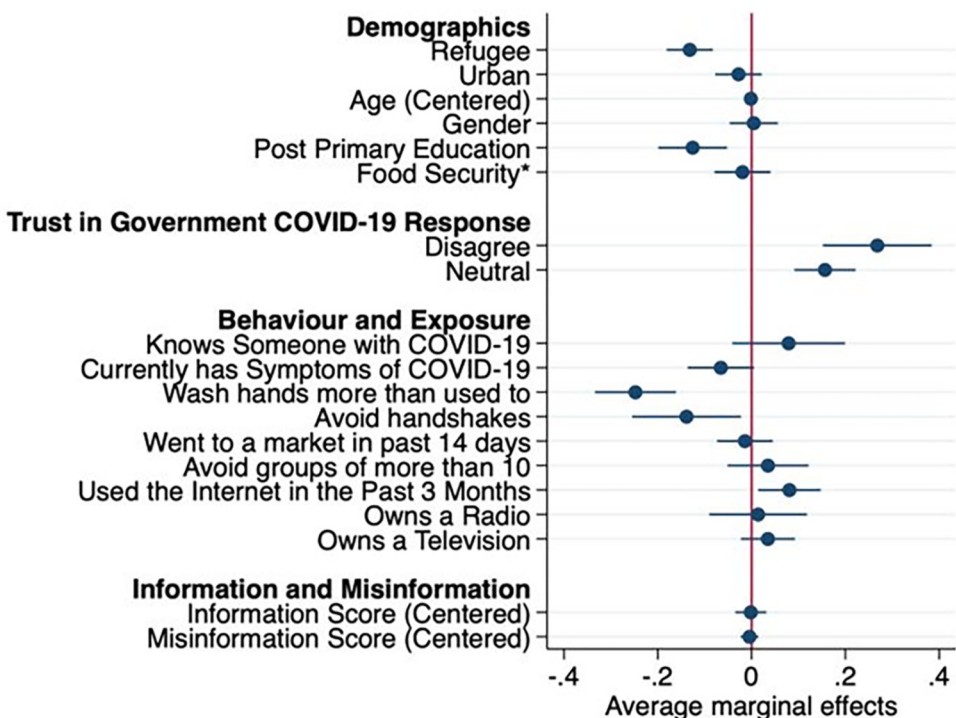

**Fig 1. Marginal effects of key variables on vaccine hesitancy, adjusted for interactions.** Marginal effects were calculated based on an adjusted logistic regression which can be found in S2 Appendix. *the exact wording of the food security question is "during the last 30 days, was there a time when you or any other adult in your household were hungry but did not eat because there was not enough money or resources for food".

hesitant that those who agreed (1.0[0.0,2.7]% vs 4.6[1.5,7.7]%); whereas non-refugees who disagreed had a higher marginal probability of being vaccine hesitant than those who agreed (35.7[23.7,47.7]% vs 7.9[5.2,10.6]%). Interactions with refugee status were also seen for knowing somebody who had COVID-19, owning a TV, and using the internet in the past three months (Table 2).

## Discussion

In our unadjusted, weighted (to the overall population of Kenya) analysis of refugees and non-refugees in Kenya, we found that 18.2% of non-refugees and 7.0% of refugees would not take the COVID-19 vaccine if offered to them at no cost. This contrasts with Orangi et al.'s (2021) much higher estimate that 4 in 10 Kenyans are vaccine hesitant (which they also define as refusal or delay of vaccination), as well as the small number of other studies from sub-Saharan Africa which also estimate higher levels of vaccine hesitancy [5, 12].

Through a weighted, adjusted analysis non-refugees in Kenya were more likely to be vaccine hesitant than refugees in Kenya. While our study is the first to directly compare vaccine hesitancy between refugees and non-refugees in a given country, Salibi et al. (2021) found that 34% of refugees over the age of fifty in Lebanon were vaccine hesitant (measured by the question "If a safe and effective vaccine for COVID-19 became available, free, would you take it) which was substantially lower than for the non-refugee population, estimated to have a vaccine hesitancy rate of 56% [13]. In addition, we found that both refugees and non-refugees educated beyond the primary level were less likely to be vaccine hesitant. In the UK, Robertson et al. (2021) estimated (for the general population) that those without any academic qualifications

**Table 2. Marginal percentages of vaccine hesitancy for interaction terms (with 95%CIs).**

| Urban | Marginal % (95%CI) |
| --- | --- |
| Non-refugee x Camp/Rural | 17.9(13.2,22.6) |
| Non-refugee x Urban | 14.9(11.1,18.7) |
| Refugee x Camp/Rural | 2.7(0.8,4.7) |
| Refugee x Urban | 6(1.1,11) |
| **Food Security** | |
| Non-refugee x No | 17.7(12.4,23) |
| Non-refugee x Yes | 15.8(11.7,20) |
| Refugee x No | 5.2(1.4,9) |
| Refugee x Yes | 1.9(0.0,4.1) |
| **Government Trust** | |
| Non-refugee x Disagree | 35.7(23.7,47.7) |
| Non-refugee x Neutral | 24.1(17.5,30.7) |
| Non-refugee x Agree | 7.9(5.2,10.6) |
| Refugee x Disagree | 1(0.0,2.7) |
| Refugee x Neutral | 3.2(-1.2,7.7) |
| Refugee x Agree | 4.6(1.5,7.7) |
| **Know Somebody who has COVID-19** | |
| Non-refugee x No | 16.4(12.7,20) |
| Non-refugee x Yes | 24.6(11.8,37.3) |
| Refugee x No | 3.8(1.1,6.4) |
| Refugee x Yes | 1.1(0.0,2.8) |
| **Used Internet in Past 3 Months** | |
| Non-refugee x No | 12.5(7.8,17.2) |
| Non-refugee x Yes | 20.9(15.5,26.3) |
| Refugee x No | 5(0.7,9.2) |
| Refugee x Yes | 2.6(0.8,4.3) |
| **Owns a TV** | |
| Non-refugee x No | 15.3(10.7,20) |
| Non-refugee x Yes | 19(14.2,23.9) |
| Refugee x No | 5(1.4,8.6) |
| Refugee x Yes | 1.3(0.0,3) |

were up to three times more likely to be vaccine hesitant than those with tertiary education (measured by the question "Imagine that a vaccine against COVID-19 was available for anyone who wanted it. How likely or unlikely would you be to take the vaccine?) [14]. We found no association between vaccine hesitancy and gender. In the UK, women and the young (aged 16–24) were more likely to be vaccine hesitant, as were female healthcare workers in Israel [14, 15]. In our study, for both refugees and non-refugees, there was no association with age, in contrast with the UK findings but in consort with the Israeli findings [15]. The lack of association with age is particularly important in Kenya, where the population distribution skews very young. This may indicate that age-specific vaccination strategies are primarily relevant in the context of groups at elevated risk for COVID-related serious illness or death (e.g., elderly) or among groups more likely to engage in behaviors placing them at greater risk of acquisition of disease (e.g., adolescents). While no evidence exists from Kenya on age-specific COVID-19-related behaviours, low levels of awareness about COVID-19 were reported among young people under the age of 25 in Mozambique, suggesting that prioritizing that populations for vaccination may be particularly helpful in interrupting the transmission of infection [16].

Our adjusted analysis also estimated that refugees who lived in camp based or rural settings were more likely to be vaccinated than their counterparts in urban settings; while non-refugees in rural settings were less likely to be vaccinated than their counterparts in urban settings. For the refugee population, we hypothesize that this is due to most rural refugees living in camps, which have a long standing and strict vaccination policy. It is possible that these refugees are simply used to conforming to vaccination requirements, which is not the case for refugees in urban settings. However, no literature has examined differences in vaccination hesitancy between refugees in different settings, and further investigation is required–something we are currently studying in Bangladesh and Kenya. For non-refugees, we estimated that those in rural settings were more likely to be vaccine hesitant than those in urban settings. This has also been estimated by other studies, including the study by Orangi et al. (2021) who found that non-refugees in rural settings were 2.5 times more likely to be vaccination hesitant than those in urban settings [5]. Among other reasons, this may be due to poor access and poor awareness [17].

The adjusted analysis also revealed that for both refugees and non-refugees hand washing and avoiding handshaking were associated with lower vaccine hesitancy. To date, no studies have examined these variables in their relationship to vaccine hesitancy. While people typically cannot avoid going to the market or eschew social contact, variables for which there was no significant association with hesitancy, they can exercise complete control over their handwashing and handshaking. It seems plausible that those practicing personal covid mitigation measures under their personal agency would also be less likely to be vaccine hesitant, perhaps through increased awareness of COVID-19. We also found that those who use the internet (as a whole, not exclusively for social media) are more likely to be vaccine hesitant, something hypothesized but not previously empirically demonstrated for the COVID-19 vaccine [18]. A plausible explanation for this is that the internet is a source of a tremendous amount of dis- and misinformation, and those who access the internet more are also relying on it more as their sole or primary source of information on the vaccine. However, this requires further investigation.

Non-refugees who did not trust the 2021 Kenyan government's COVID-19 response were more likely to be vaccine hesitant, as suggested by Afolabi et al. (2021) (who define vaccine hesitancy as the delay or blunt refusal of vaccines) [19]. However, this was not the case among refugees: refugees who did trust the 2021 Kenyan government's COVID-19 response were more likely to be vaccine hesitant than those who do not. These counterintuitive results have also been seen in high income countries. Trent et al. (2021) report that government trust was associated with an increased likelihood of vaccination in two Australian cities, but a decreased likelihood of vaccination in two American cities [20]. The authors report that this may be due to the politization of the vaccine in the US, with the government in power at the time in the US substantially contributing to misinformation [20]. Among the refugee population in Kenya however, the reason for this is unclear and requires further investigation, particularly in context of the views of the refugees on the government's attitudes towards science and public health. Perhaps there are similar issues with the refugees responding to their trust in the government of the country of origin; or perhaps there may be fear in stating government distrust. Alternatively, mistrust may result in conformity to government regulations such as COVID-19 vaccination, due to fear of retaliation by the government; or a decrease in personal risk perception through trusting of government policies.

This study has several limitations: First, the sample consisted of households who had access to a phone and answered it, though the weights were designed to account for this. Second, the sample omitted participants who did not answer some of the key questions, though this number was minimal; Third, important confounders, such as religion, income, and ethnic

background were not available in the dataset, which should be explored in subsequent studies. Fourth, as in any survey, people may not have answered truthfully if they feared consequences for their answers or were uncomfortable answering in a specific way, particularly the refugee population. While we do anticipate that assurances of privacy partially protected against this, further work must be conducted to understand the extent of misreporting. Finally, due to the sampling strategy, unregistered refugees were not approached, possibly representing a particularly vulnerable group that differs in systematic ways from registered refugees. Despite these limitations, this is a large and diverse population-based sample that explores the understudied area of COVID-19 vaccine hesitancy among refugees and compares them with non-refugee populations in the host country.

## Conclusions

We found, in Kenya, that refugees differed on several key several aspects from non-refugees with regard to COVID-19 vaccine hesitancy. The findings of this study suggest that while some factors in vaccine hesitancy are similar between refugees and non-refugees (education, internet use, and COVID mitigation measures), other factors differ. These differing factors call for differentiated programming. Further, future research on vaccine hesitancy is needed to elucidate the impact of religion, ethnic background, income, and other factors; and factors related to not returning for the second dose or booster of the COVID-19 vaccination, a particular problem in Kenya [21]. Research should also be taken into the efficacy of possible interventions. Caution should be used however in using our findings in other countries, where separate analyses should be conducted as these results may not be generalizable.

## Supporting information

**S1 Appendix. Scaling of information and misinformation variables to create information and misinformation scores for the logistic regression model.**
(DOCX)

**S2 Appendix. A weighted, adjusted multivariable logistic regression showing odds of being vaccine hesitant among the population of Kenya.**
(DOCX)

**S1 Questionnaire.**
(DOCX)

## Acknowledgments

**Disclaimers:** The views expressed in this paper are those of the authors and do not necessarily reflect those of the World Bank.

## Author Contributions

**Conceptualization:** Ryan T. Rego, Akbar K. Waljee.

**Data curation:** Ryan T. Rego, Antonia Johanna Sophie Delius.

**Formal analysis:** Ryan T. Rego, Eileen Weinheimer-Haus, Ji Zhu, Timothy Hofer, Akbar K. Waljee.

**Investigation:** Ryan T. Rego, Timothy Hofer, Akbar K. Waljee.

**Methodology:** Ryan T. Rego, Anthony K. Ngugi, Timothy Hofer, Akbar K. Waljee.

**Supervision:** Joseph C. Kolars, Akbar K. Waljee.

**Writing – original draft:** Ryan T. Rego, Anthony K. Ngugi, Antonia Johanna Sophie Delius, Stanley Luchters, Joseph C. Kolars, Furqan B. Irfan, Eileen Weinheimer-Haus, Amina Abubakar, Ji Zhu, Timothy Hofer, Akbar K. Waljee.

**Writing – review & editing:** Ryan T. Rego, Anthony K. Ngugi, Antonia Johanna Sophie Delius, Stanley Luchters, Joseph C. Kolars, Furqan B. Irfan, Eileen Weinheimer-Haus, Amina Abubakar, Reena Shah, Ji Zhu, Matthew L. Boulton, Timothy Hofer, Akbar K. Waljee.

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
