## [Decision Letter · Decision Letter 0]

7 Feb 2022

PGPH-D-21-00986

Factors Associated with COVID-19 Vaccine Hesitancy Among Nationals and Refugees in Kenya

Dear Dr. Ryan Trevor Titus Rego,

Thank you for submitting your manuscript to PLOS Global Public Health. After careful consideration, we feel that it has merit but does not fully meet PLOS Global Public Health’s publication criteria as it currently stands. Therefore, we invite you to submit a revised version of the manuscript that addresses the points raised during the review process.

We look forward to receiving your revised manuscript.

Kind regards,

Muhammad Asaduzzaman, MD MPH MPhil

Academic Editor

Journal Requirements:

1. Please include a complete copy of PLOS’ questionnaire on inclusivity in global research in your revised manuscript. Our policy for research in this area aims to improve transparency in the reporting of research performed outside of researchers’ own country or community. The policy applies to researchers who have travelled to a different country to conduct research, research with Indigenous populations or their lands, and research on cultural artefacts. The questionnaire can also be requested at the journal’s discretion for any other submissions, even if these conditions are not met.  Please find more information on the policy and a link to download a blank copy of the questionnaire here: https://journals.plos.org/plosone/s/best-practices-in-research-reporting. Please upload a completed version of your questionnaire as Supporting Information when you resubmit your manuscript.

2. Please provide  separate figure files in .tif or .eps format only and remove any figures embedded in your manuscript file.  Please ensure that all files are under our size limit of 20MB.  

For more information about how to convert your figure files please see our guidelines: Once you've converted your files to .tif or .eps, please also make sure that your figures meet our format requirements

Additional Editor Comments (if provided):

Thanks for submission of an important topic in the current pandemic. However, the manuscript requires major revision in the article structure formatting, study design and more discussion with literature review. Currently, it looks like a 'Letter to Editor' or 'Rapid Communication' article. Please follow the author instructions of the journal and comments of the reviewers.

Reviewers' comments:

Reviewer's Responses to Questions

**Comments to the Author**

1. Does this manuscript meet PLOS Global Public Health’s publication criteria? Is the manuscript technically sound, and do the data support the conclusions? The manuscript must describe methodologically and ethically rigorous research with conclusions that are appropriately drawn based on the data presented.

Reviewer #1: Yes

Reviewer #2: Partly

2. Has the statistical analysis been performed appropriately and rigorously?

Reviewer #1: Yes

Reviewer #2: Yes

3. Have the authors made all data underlying the findings in their manuscript fully available (please refer to the Data Availability Statement at the start of the manuscript PDF file)?

Reviewer #1: Yes

Reviewer #2: Yes

4. Is the manuscript presented in an intelligible fashion and written in standard English?

Reviewer #1: Yes

Reviewer #2: Yes

5. Review Comments to the Author

Reviewer #1: I read with great interest the paper that is on an important issue from interesting setting. But need major revision.

Below my suggestions

1. Introduction: updata data on SARS CoV2 wordwilde. Furthermore introduce better the role of SARS CoV2 in your country and also as impact in health services. We know, from others experiences, how pandemic reduce the health services also for re-allocation of heatlworker and disruption of other services (for tb, malaria or maternal services), Introduce also this aspect as context. (see and cite Quaglio G, Tognon F, et al. Impact of Ebola outbreak on reproductive health services in a rural district of Sierra Leone: a prospective observational study. BMJ Open. 2019 Sep 4;9(9):e029093. doi: 10.1136/bmjopen-2019-029093. )

2. Methods and result: are clear

3. Discussion: discuss better and compare your results with other paper. Furthermore, the role of young is crucial in adherence in vaccination axpecially in Africa context where median age of population is young. (see and cite Marotta C, Nacareia U et al. Mozambican Adolescents and Youths during the COVID-19 Pandemic: Knowledge and Awareness Gaps in the Provinces of Sofala and Tete. Healthcare (Basel). 2021 Mar 13;9(3):321. doi: 10.3390/healthcare9030321. )

Furthermore, give some global health proposal that came from your paper

Reviewer #2: The data analysis of this study is strong. However, the introduction is superficial and does not provide an acceptable rationale for the study of vaccine hesitancy among refugees in Kenya. The authors do not present the picture of the pandemic in Kenya. No information on Covid-19 vaccine update is provided despite over 10.5 million Covid-19 vaccine doses administered in Kenya. The main finding from the study and its implication are not discussed adequately. Conclusion section is inadequate. I have included additional major comments in the attached PDF.

6. PLOS authors have the option to publish the peer review history of their article (what does this mean?). If published, this will include your full peer review and any attached files.

**Do you want your identity to be public for this peer review?** For information about this choice, including consent withdrawal, please see our Privacy Policy.

Reviewer #1: **Yes: **Francesco Di Gennaro

Reviewer #2: No

---

## [Decision Letter · Decision Letter 1]

7 Jun 2022

PGPH-D-21-00986R1

Factors Associated with COVID-19 Vaccine Hesitancy Among Refugees and Non-Refugees in Kenya

Dear Dr. Ryan Trevor,

Thank you for submitting your manuscript to PLOS Global Public Health. After careful consideration, we feel that it has merit but does not fully meet PLOS Global Public Health’s publication criteria as it currently stands. Therefore, we invite you to submit a revised version of the manuscript that addresses the points raised during the review process.

We look forward to receiving your revised manuscript.

Kind regards,

Muhammad Asaduzzaman, MD MPH MPhil

Academic Editor

Journal Requirements:

1. Please amend your Financial Disclosure statement. If you did not receive any funding for this study, please simply state: “The authors received no specific funding for this work.”

2. Please update your Competing Interests statement. If you have no competing interests to declare, please state: “The authors have declared that no competing interests exist.”

3. Please provide a complete Data Availability Statement in the submission form. If your research concerns only data provided within your submission, please write “All data are in the manuscript and/or supporting information files.” as your Data Availability Statement.

4. Please include a separate legend for Figure 1 in your manuscript.

Additional Editor Comments (if provided):

Dear authors

Thanks a lot for the revised version. It has been improved a lot though still needs some changes suggested by the reviewer. Hope to see the amended version soon. Best of luck.

Reviewers' comments:

Reviewer's Responses to Questions

**Comments to the Author**

1. If the authors have adequately addressed your comments raised in a previous round of review and you feel that this manuscript is now acceptable for publication, you may indicate that here to bypass the “Comments to the Author” section, enter your conflict of interest statement in the “Confidential to Editor” section, and submit your "Accept" recommendation.

Reviewer #1: All comments have been addressed

Reviewer #2: (No Response)

2. Does this manuscript meet PLOS Global Public Health’s publication criteria? Is the manuscript technically sound, and do the data support the conclusions? The manuscript must describe methodologically and ethically rigorous research with conclusions that are appropriately drawn based on the data presented.

Reviewer #1: Yes

Reviewer #2: Partly

3. Has the statistical analysis been performed appropriately and rigorously?

Reviewer #1: Yes

Reviewer #2: I don't know

4. Have the authors made all data underlying the findings in their manuscript fully available (please refer to the Data Availability Statement at the start of the manuscript PDF file)?

Reviewer #1: Yes

Reviewer #2: No

5. Is the manuscript presented in an intelligible fashion and written in standard English?

Reviewer #1: Yes

Reviewer #2: Yes

6. Review Comments to the Author

Reviewer #1: The authors improved their paper that now can be accept

Reviewer #2: GENERAL COMMENTS

This revised version of the manuscript represents an improvement over the initial submission. The authors have addressed several of my comments. However, given the important number of authors on this paper and the multiple questions raised in this revised manuscript, I wanted to see a statement about author’s contribution, but it was not available. Apparently, the list of authors has been expanded from 11 in the original submission to 13 in the revised manuscript. In total, the authorship combines 14 PhDs and MDs. Given this amount of experience and intellectual asset, I strongly believe that this paper can be much better if all the authors carefully read it and contribute meaningfully to it. I recommend that all the authors attend to the outstanding issues listed in the attached file.

My detailed comments for each section of the paper are included in the attached file.

7. PLOS authors have the option to publish the peer review history of their article (what does this mean?). If published, this will include your full peer review and any attached files.

**Do you want your identity to be public for this peer review?** For information about this choice, including consent withdrawal, please see our Privacy Policy.

Reviewer #1: **Yes: **Francesco Di Gennaro

Reviewer #2: No

---

## [Editor Report · Decision Letter 2]

26 Jul 2022

Factors Associated with COVID-19 Vaccine Hesitancy Among Refugees and Non-Refugees in Kenya

PGPH-D-21-00986R2

Dear Dr. Ryan Trevor Titus Rego,

We are pleased to inform you that your manuscript 'Factors Associated with COVID-19 Vaccine Hesitancy Among Refugees and Non-Refugees in Kenya' has been provisionally accepted for publication in PLOS Global Public Health.

Best regards,

Muhammad Asaduzzaman, MD MPH MPhil

Academic Editor
